# 11% efficiency solid-state dye-sensitized solar cells with copper(II/I) hole transport materials

Yiming Cao[1,*], Yasemin Saygili[2,*], Amita Ummadisingu[1], Joël Teuscher[3], Jingshan Luo[1], Norman Pellet[1], Fabrizio Giordano[1], Shaik Mohammed Zakeeruddin[1], Jacques-E. Moser[3], Marina Freitag[2,†], Anders Hagfeldt[2] & Michael Grätzel[1]

Solid-state dye-sensitized solar cells currently suffer from issues such as inadequate nano-pore filling, low conductivity and crystallization of hole-transport materials infiltrated in the mesoscopic $TiO_2$ scaffolds, leading to low performances. Here we report a record 11% stable solid-state dye-sensitized solar cell under standard air mass 1.5 global using a hole-transport material composed of a blend of [Cu (4,4′,6,6′-tetramethyl-2,2′-bipyridine)$_2$](bis(trifluoromethylsulfonyl)imide)$_2$ and [Cu (4,4′,6,6′-tetramethyl-2,2′-bipyridine)$_2$](bis(trifluoromethylsulfonyl)imide). The amorphous Cu(II/I) conductors that conduct holes by rapid hopping infiltrated in a 6.5 μm-thick mesoscopic $TiO_2$ scaffold are crucial for achieving such high efficiency. Using time-resolved laser photolysis, we determine the time constants for electron injection from the photoexcited sensitizers Y123 into the $TiO_2$ and regeneration of the Y123 by Cu(I) to be 25 ps and 3.2 μs, respectively. Our work will foster the development of low-cost solid-state photovoltaic based on transition metal complexes as hole conductors.

[1] Laboratory of Photonics and Interfaces, Institute of Chemical Sciences & Engineering, École Polytechnique Fédérale de Lausanne, 1015 Lausanne, Switzerland. [2] Laboratory of Photomolecular Science, Institute of Chemical Sciences & Engineering, École Polytechnique Fédérale de Lausanne, 1015 Lausanne, Switzerland. [3] Photochemical Dynamics Group, Institute of Chemical Sciences & Engineering, École Polytechnique Fédérale de Lausanne, 1015 Lausanne, Switzerland. * These authors contributed equally to this work. † Present address: Department of Chemistry, Ångström Laboratory, Uppsala University, 751 20 Uppsala, Sweden. Correspondence and requests for materials should be addressed to M.F. (email: marina.freitag@kemi.uu.se) or to A.H. (email: anders.hagfeldt@epfl.ch) or to M.G. (email: michael.graetzel@epfl.ch).

Natural systems have cleverly used copper complexes as efficient electron-transfer mediators by constraining them in the protein matrix to minimize the structural change between copper(II) and copper(I), which results in a relatively small internal reorganization barrier to electron transfer[1–4]. Several copper model complexes, that is, bis(1,10-phenanthroline)copper ($[Cu(phen)_2]^{2+/+}$), $[(-)$-sparteine-$N,N']$(maleonitriledithiolato-$S,S'$)copper ($[Cu(SP)(mmt)]^{0/-}$) and bis(2,9-dimethy-1,10-phenanthroline)copper ($[Cu(dmp)_2]^{2+/+}$), were first demonstrated as fast electron-transfer mediators in electrolytes for solar-to-electricity conversion in dye-sensitized solar cells (DSCs)[5]. Among these complexes, the sterically constrained $[Cu(dmp)_2]^{2+/+}$ has exhibited the fastest electron self-exchange rate ($23 M^{-1} s^{-1}$) and the highest power conversion efficiency (PCE) of 2.2% under a light intensity of $200 W m^{-2}$.

Profiting from the rapid electron self-exchange rate in rigid copper complexes, $[Cu(dmp)_2]^{2+/+}$ molecules were recently introduced as a hole-transport material (HTM) for solid-state DSCs (ssDSCs)[6]. The so called 'zombie' ssDSCs were made simply by evaporating volatile solvents from the $[Cu(dmp)_2]^{2+/+}$ redox shuttle electrolyte in ambient air. Surprisingly, the ssDSC showed a high short-circuit photocurrent density ($J_{sc}$) of $13.8 mA cm^{-2}$, exceeding the $J_{sc}$ of a liquid electrolyte-based DSC ($9.4 mA cm^{-2}$). The PCE of the ssDSC was 8.2% under standard air mass 1.5 global (AM1.5G) conditions, a performance superior to those of counterparts made using copper(I) iodide (4.5%)[7], copper(I) thiocyanate (2%)[8], poly(3,4-ethylenedioxythiophene) (PEDOT) (7.1%)[9], 2,2',7,7'-tetrakis($N,N$-di-p-methoxyphenylamine)-9,9'-spirobifluorene (spiro-MeOTAD) (7.7%)[10], $Cs_2SnI_6$ (7.8%)[11] or cobalt complexes[12] as HTMs.

Here we show a record PCE of 11.0% (average 10.2%) for stable ssDSC under standard AM1.5G conditions, fabricated using a blend of $[Cu(tmby)_2](TFSI)_2$ and $[Cu(tmby)_2](TFSI)$ (tmby or tmbp[13] $= 4,4',6,6'$-tetramethyl-2,2'-bipyridine; TFSI $=$ bis(trifluoromethylsulfonyl)imide) as a HTM (Fig. 1a), Y123 as a sensitizer and electrodeposited PEDOT as a counter electrode. Substituent methyl groups at the 6,6'-positions provide a rigid framework for the Cu(II) and Cu(I) complexes[13] and reduce the structural changes between them to allow for rapid self-exchange electron transfer[14]. We investigate the infiltration of the copper complexes in a 6.5 μm-thick mesoscopic $TiO_2$ film, the conductivity of the HTM, the influence of crystallinity of the hole conductor and the photo-induced interfacial charge transfer dynamics of Y123 dye molecules in our Cu(II/I) HTM-based ssDSCs.

## Results

**Cu(II/I) complexes for ssDSCs**. $[Cu(tmby)_2](TFSI)_2$ and $[Cu(tmby)_2](TFSI)$ have maximum molar absorption coefficients of $1,400 M^{-1} cm^{-1}$ at 360 nm and $5,300 M^{-1} cm^{-1}$ at 451 nm, respectively[15], inferior to that of the organic dye Y123 (around $50,000 M^{-1} cm^{-1}$ at 530 nm)[16]. A substantial fraction of the solar photons are thus harvested by the dye molecules for charge generation. Furthermore, the energetic alignment of the redox potential of $[Cu(tmby)_2](TFSI)$ (0.87 V versus standard hydrogen electrode[15]) with that of Y123 (1.07 V versus standard hydrogen electrode[15]) allows efficient dye regeneration following the initial photo-induced interfacial charge separation step. We fabricate the DSCs by adsorbing Y123 as the sensitizer onto the $TiO_2$ surface, employing $[Cu(tmby)_2]^{2+/+}$ as the redox shuttle in a volatile electrolyte and electrodepositing PEDOT on fluorine-doped tin oxide (FTO) as the counter electrode. The electrolyte contains 0.06 M $[Cu(tmby)_2](TFSI)_2$, 0.2 M $[Cu(tmby)_2](TFSI)$, 0.1 M LiTFSI and 0.6 M 4-*tert*-butylpyridine (TBP) in acetonitrile (ACN). DSCs with this electrolyte show a PCE of 9.3 ± 0.3%,

with open-circuit photovoltage $V_{oc} = 1.07 ± 0.03$ V, $J_{sc} = 13.17 ± 0.46$ mA cm$^{-2}$ and fill factor (FF) $= 0.66 ± 0.02$ under standard AM1.5G conditions (red symbols and error bars in Fig. 1b–e), a value comparable to previous results[15]. As shown in Supplementary Fig. 1, the high $V_{oc}$ of Cu(II/I)-based DSCs stems from a large energy gap between the conduction band edge of $TiO_2$ and redox potential of the electrolyte, together with a small reorganization energy for the interfacial electron-transfer reaction from the Cu(I) complex to the oxidized sensitizer affording rapid dye regeneration at a minimal energy expenditure[15].

Here our investigation focuses on ssDSCs employing a blend of $[Cu(tmby)_2]^{2+}$ and $[Cu(tmby)_2]^{+}$ as solid-state HTM. We produced the devices by slowly evaporating the solvents of the electrolyte in ambient air through the unsealed hole on the counter electrode[6]. Fortunately, the methyl group substituents at the 6,6'-positions can stabilize the copper complexes against oxidation and moisture[17]. The evaporation of ACN induces the solidification of $[Cu(tmby)_2]^{2+/+}$, facilitated by the low solubility of the copper complexes in this solvent. The residual solid copper complexes contain small amounts of entrapped ACN, which has negligible effects on the device performance. We note that a slow evaporation of solvents is pivotal to yield efficient solar cells, because rapid evaporation leads to poor contact between the solid-state crystalline HTM and the electrodes (Supplementary Fig. 2). Figure 1b–e shows the evolution of the photovoltaic performance of the unsealed DSCs. Strikingly, the average PCE of unsealed DSCs increases to over 10% after 5 days in ambient air due to improvements in the $J_{sc}$, $V_{oc}$ and FF.

Figure 2a shows the cross-sectional scanning electron microscopy (SEM) image of the mesoscopic-sensitized $TiO_2$ scaffold infiltrated by the solid Cu(II/I) HTM after the 5 days drying period and the magnified SEM images are presented in Supplementary Fig. 3. The solid Cu(II/I) HTM layer covering the mesoscopic $TiO_2$ film has a thickness of about 2.0 μm. Apparently rapid hole transport can occur across this layer resulting in a high FF for the device. The thickness of the overlayer can be controlled by tuning the thickness of thermoplastic spacer between the electrodes. From energy dispersive X-ray spectroscopy analysis, we show that the solid Cu(II/I) HTM is homogeneously infiltrated in the 6.5 μm-thick sensitized mesoporous $TiO_2$ scaffold (Supplementary Fig. 4). Unlike our HTM, using spiro-MeOTAD as HTM for ssDSCs[18] generally calls for thinner mesoscopic $TiO_2$ films (<3 μm) to ensure maximum filling in mesopores[19]. However, the light harvesting efficiency suffers from reducing thickness of the mesoscopic $TiO_2$ films.

**Rapid hole transport in solid Cu(II/I) HTM**. To study the rapid hole hopping in the solid Cu(II/I) HTM for ssDSCs, we performed electrical impedance spectroscopy (EIS) measurements under light illumination. We observe that the conductivity in solid Cu(II/I) HTM of the ssDSCs is higher than that in the volatile electrolyte of the DSCs under a photovoltage (Fig. 2b) or a photocurrent (Supplementary Fig. 5), indicating a rapid hole hopping in solid Cu(II/I) HTM. Remarkably, the conductivity of our Cu(II/I) HTM is more than 10 times higher than that of spiro-MeOTAD in a mesoscopic $TiO_2$ film at a light intensity of $1,000 W m^{-2}$ conditions[20]. It was shown that the HTM conductivity can limit the charge transport in DSCs when the carrier density in $TiO_2$ is higher than $2 \times 10^{17} cm^{-3}$ at short-circuit conditions[21]. In our solar cells under full sun illumination, the carrier density of $TiO_2$ is over $7 \times 10^{17} cm^{-3}$ at short-circuit conditions measured by charge extraction methods[22]. Therefore, the increased conductivity of Cu(II/I) HTM contributes to an improvement in the $J_{sc}$ of ssDSCs. Moreover, there is little

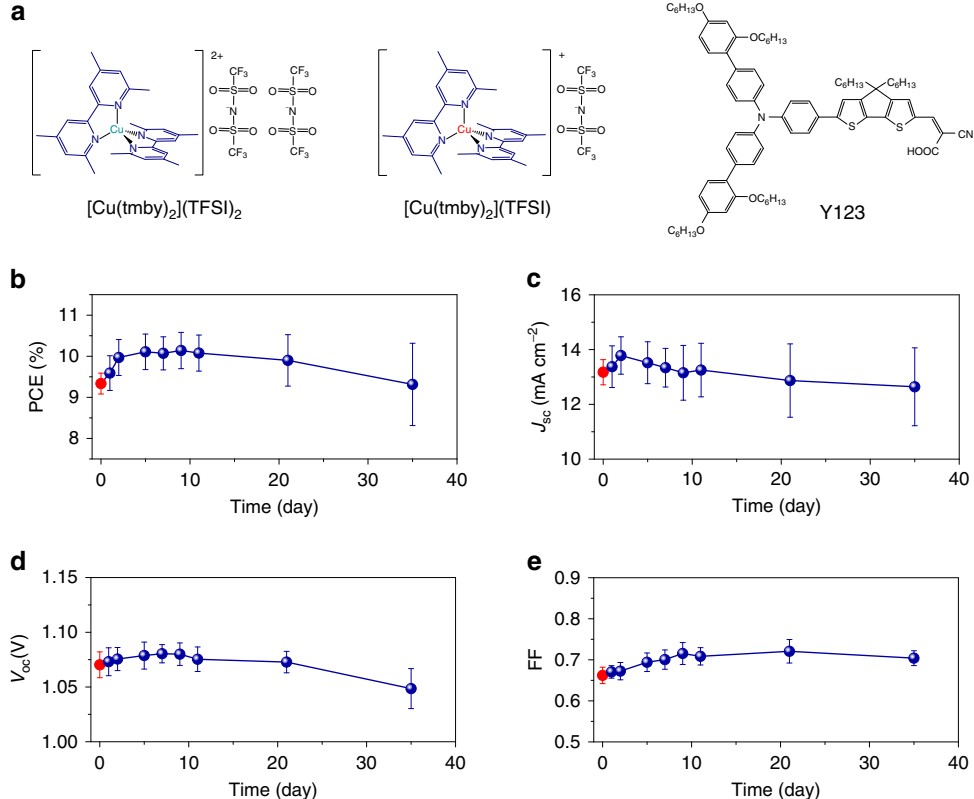

**Figure 1 | DSCs with the Cu(II/I) hole conductor and Y123 sensitizer.** (**a**) Schematic structures of the hole conductors based on [Cu(4,4′,6,6′-tetramethyl-2,2′-bipyridine)$_2$](bis(trifluoromethylsulfonyl)imide)$_2$ ([Cu(tmby)$_2$](TFSI)$_2$) and [Cu(4,4′,6,6′-tetramethyl-2,2′-bipyridine)$_2$](bis(trifluoromethylsulfonyl)imide), ([Cu(tmby)$_2$](TFSI)) and the dye molecule Y123. (**b**) Evolution of the PCE of unsealed DSCs. (**c**) Evolution of the $J_{sc}$ of unsealed DSCs. (**d**) Evolution of the $V_{oc}$ of unsealed DSCs. (**e**) Evolution of the FF of the unsealed DSCs. The unsealed DSCs were stored in ambient air in the dark. The photovoltaic parameters of solar cells were measured under standard AM1.5G conditions. The red symbols and error bars denote the sealed DSCs based on the volatile electrolyte. The royal blue symbols and error bars denote unsealed DSCs. The average (symbols) and s.d. (error bars) were calculated from solar cells numbering between 4 and 22.

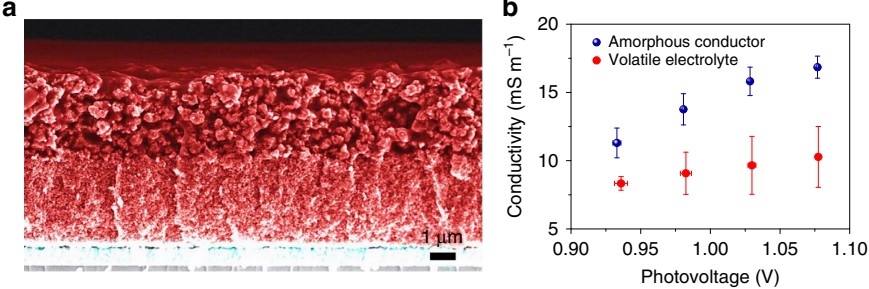

**Figure 2 | Solid Cu(II/I) hole conductor in solid-state DSCs.** (**a**) Cross-sectional SEM image of a solid-state DSC without the counter electrode. Moving from the bottom to the top, layers of compact TiO$_2$-coated FTO (cyan), 3.5 μm-thick transparent mesoscopic TiO$_2$ + 3.0 μm-thick light scattering TiO$_2$ (red), and 2.0 μm-thick solid Cu(I/II) hole conductor overlayer (red) are visible. (**b**) Photovoltage-dependent conductivity in the solid hole conductor and the volatile electrolyte of solar cells as obtained from the EIS analysis. The solar cells were measured under illumination at 1,000 W m$^{-2}$ supplied by a white light-emitting diode. The average (symbols) and s.d. (error bars) were calculated from solar cells numbering between four and six.

difference in the light harvesting capacity between the DSCs based on the solid and liquid hole conductors (Supplementary Fig. 6).

In striking contrast we observe a dramatic loss of performance on slow solvent evaporation from a volatile electrolyte containing the tris(bipyridine) cobalt(III/II) complexes as a redox couple. The PCE of the DSC drops sharply from 9.8 to 0.1% (Supplementary Fig. 7). The reduction in $J_{sc}$ from 14.2 to 0.2 mA cm$^{-2}$ is partly due to the extremely sluggish outer sphere electron self-exchange between the Co(III) and Co(II) complexes (0.645 M$^{-1}$ s$^{-1}$)[23]. Although tris(bipyridine) cobalt(III/II) containing volatile electrolytes have

pushed the PCE of DSCs from 6.7 (ref. 24) to over 13% (refs 25–27), the encapsulation of such highly efficient DSCs to prevent solvent leakage remains a challenge. Recently, ssDSCs using a blend of [Co(bpyPY4)](OTf)$_3$ and [Co(bpyPY4)](OTf)$_2$ (bpyPY4 = hexadentate ligand 6,6′-bis(1,1-di(pyridin-2-yl)ethyl)-2,2′-bipyridine; OTf = trifluoromethanesulfonate) as a HTM gave a PCE of 5.7% under standard AM1.5G[12]. However, this ssDSC had a lower $J_{sc}$ (12.12 mA cm$^{-2}$) and PCE than the liquid counterpart (17.2 mA cm$^{-2}$ and 9.6%), which is not the case when our copper complexes with a small internal reorganization energy[14,15] are used.

**Crystallinity of solid Cu(II/I) HTM**. Our ssDSCs stored in ambient air for over 20 days have large variations in $J_{sc}$ as shown in Fig. 1c, because a few solar cells display a sublinear dependence of the $J_{sc}$ on light intensity ($P$) according to the power law[28], $J_{sc} \propto P^{\alpha}$, where $\alpha < 1$ is the exponent. Such a ssDSC has an $\alpha$ of 0.81 as shown in Fig. 3a, while the most efficient one has an $\alpha$ of 0.99. To probe the molecular origin of this observation, we performed XRD measurements on the samples of sensitized TiO$_2$ films combined with the Cu(II/I) HTM in both kinds of ssDSCs. Apart from reflections corresponding to FTO and TiO$_2$ in Fig. 3b, we see reflections (at $2\theta$ values of 16.0° and 16.6°) in the sample for a ssDSC with a sublinear dependence of the $J_{sc}$, pointing to the presence of a crystalline phase within the Cu(II/I) HTM. The absence of these reflections in the sample for an ssDSC with a linear dependence of the $J_{sc}$ indicates that the HTM is amorphous in this case. The holes in the crystalline HTM are likely to get trapped at crystal grain boundaries, leading to reduced hole mobility and a built-up space charge layer in the solar cells as a consequence of the difference in electron and hole mobilities[29].

Compared to the amorphous Cu(II/I) HTM, the crystalline one displays a slightly blue shifted peak in the absorption spectrum and enhanced absorption in the metal-to-ligand charge transfer transitions region, as shown in Fig. 3c. This result highlights the magnitude of structural difference between the ground and excited states. We infer that copper complexes in the crystalline HTM adopt a flattened distorted tetrahedral structure by reducing the dihedral angle between the ligands[30–32], while in the amorphous state they are less flattened. A similar flattened distorted structure that intensifies the absorption in the weak broad metal-to-ligand charge transfer region was also observed in crystals of [Cu(dpphen)$_2$]$^+$ (dpphen = 2,9-diphenyl-1,10-phenanthroline)[33].

We note that the aggregation of copper complexes by evaporation of solvents in the electrolyte enhances the photoluminescence of the HTM. The crystalline HTM has a stronger emission than the amorphous one. We posit that the photoluminescence of the HTM originates from the Cu(I), because the solid Cu(II) film shows a negligible emission at 670 nm (Supplementary Fig. 8). We performed nanosecond photoluminescence decay dynamics measurements of the amorphous and crystalline HTMs. After deconvolution of the instrument response function, the traces of the amorphous hole conductor in Fig. 3d are best fitted by a bi-exponential decay function, giving photoluminescence decay time components (and pre-exponential factors) of 94 ps (0.78) and 2.0 ns (0.22), while the traces of the crystalline sample in Fig. 3e are fitted by the bi-exponential decay function, giving 2.1 ns (0. 96) and 37.9 ns (0.04). The longer photoluminescence decay times in the crystalline HTM support a flattened distorted structure of the copper complexes[32–34]. The crystalline phase also influences the molecular photochemistry[35], for instance, copper complexes in the crystalline state are quite restricted by the closely packed molecules, resulting in longer photoluminescence decay times.

**Interfacial charge transfer dynamics in Cu(II/I)-based ssDSC.** We performed picosecond time-resolved transient absorption spectroscopy to investigate the electron injection dynamics of Y123. We selected 650 nm as a suitable wavelength to probe for electron injection, because we observed a transient absorbance change of the photoexcited Y123 specie on an Al$_2$O$_3$ film, while the absorbance of the photo-oxidized Y123 on a TiO$_2$ film does not differ from the dye ground state absorbance at this wavelength (Supplementary Fig. 9). Moreover, exclusive monitoring of excited dye eliminates charge accumulation effects that could

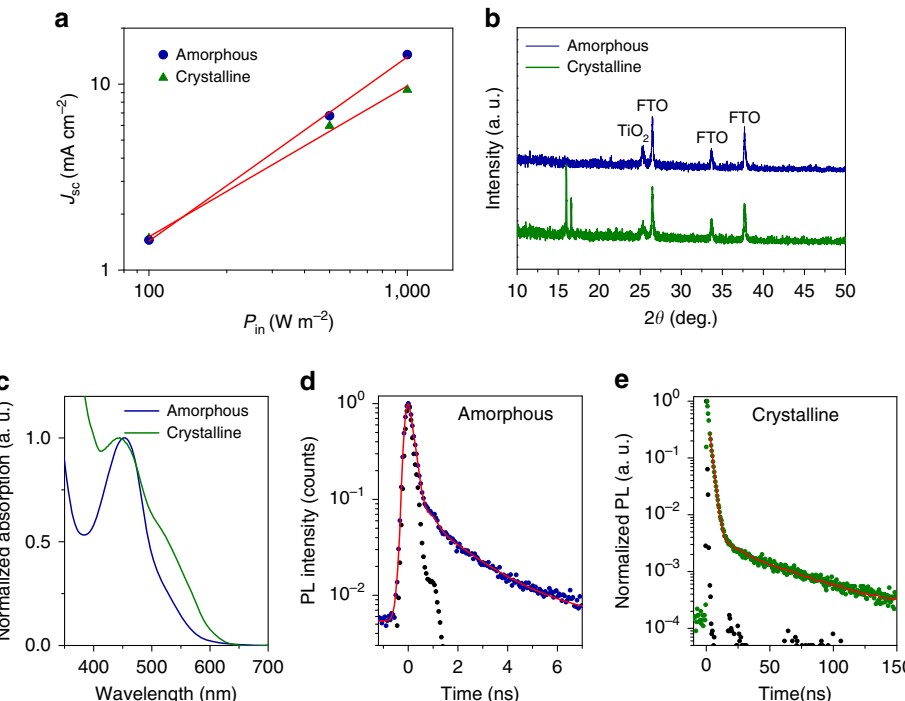

**Figure 3 | Comparison of amorphous and crystalline Cu(II/I) hole conductor. (a)** Incident light intensity ($P_{in}$)-dependent $J_{sc}$ of solid-state DSCs with amorphous and crystalline hole conductors. The red lines are linear fits of the data. **(b)** X-ray diffraction of amorphous and crystalline Cu(II/I) hole conductors in conjunction with Y123-sensitized TiO$_2$ film on FTO glass substrate. **(c)** Absorption spectra for the amorphous and crystalline Cu(II/I) hole conductors. **(d)** Photoluminescence decay dynamics of amorphous Cu(II/I) hole conductor. **(e)** Photoluminescence decay dynamics of crystalline Cu(II/I) hole conductor. The traces were collected at 670 nm with photoluminescence maximum following nanosecond laser pulsed excitation at 408 nm. The black dot symbols are instrument response function. The red lines are bi-exponential fits of the data.

arise from the 6 ms lifetime of oxidized Y123, when probed at 1 kHz. However, charge accumulation certainly occurs and the experiment is possibly performed at higher charge density than suggested by the fluence only.

Figure 4a shows two distinct dynamic events at 650 nm for the sample of $Al_2O_3$/Y123/inert electrolyte (0.1 M LiTFSI and 0.6 M TBP in ACN): an initial fast rise with a 1 ps time constant ($\tau$) followed by a monoexponential decay ($\tau = 337$ ps). The initial fast component is likely to arise from an excited state species formed by an intramolecular charge separation process forming an intramolecular radical ion pair following dye excitation[36]. The slow component is attributed to the lifetime of this excited species. On replacing $Al_2O_3$ film by $TiO_2$, the sample of $TiO_2$/Y123/inert electrolyte shows an initial fast rise followed by a monoexponential decay with a $\tau$ of 12 ps, attributed to the electron injection from the photoexcited Y123. Therefore, electron injection is in competition with the intramolecular process, indicating that it occurs from these different excited species. Because the signal recovers to zero at 650 nm, electron injection is unambiguously characterized and no event can occur at a longer timescale. On combining $TiO_2$/Y123 with the solid Cu(II/I) HTM, we observe a variation in the electron injection lifetime with a $\tau$ of 25 ps, which is longer than that of $TiO_2$/Y123 without the HTM (12 ps). This result is probably due to the effects of the environment, because a longer lifetime constant of 750 ps of photoexcited Y123 on $Al_2O_3$ with the HTM is also observed. Overall, we extract electron injection yields of 97% from these measurements. Our result differs from the previous time constant of 2 ps for the electron injection from Y123 in samples composed of large $TiO_2$ nanoparticles and cobalt complexes-based electrolyte, measured using the diffuse reflectance spectroscopy method[37].

The electron donating dynamics of Cu(I) to the photo-oxidized Y123 was resolved by nanosecond transient absorption spectroscopy. The photo-oxidized Y123 species on the $TiO_2$ surface have a strong absorption at 715 nm (ref. 15), at which wavelength the transient absorption traces were collected as shown in Fig. 4b. In the sample of $TiO_2$/Y123/inert electrolyte, photo-oxidized Y123 species recombine with injected electrons in the $TiO_2$ film. The time constant for this process is determined to be 6.0 ms. In the sample of $TiO_2$/Y123/solid Cu(II/I) HTM, the photo-oxidized Y123 species either recombine with injected electrons from $TiO_2$ or are regenerated by the Cu(I) in the HTM. The time constant for the decay traces of this sample is 3.2 µs. This result indicates that the electron donation from Cu(I) to photo-oxidized Y123 outcompetes charge recombination by a factor of 938 ascertaining-near quantitative regeneration of the sensitizer.

**Cu(II/I) HTM for record efficiency ssDSC.** Figure 5a shows the histogram of PCEs for 52 ssDSCs using the blend of $[Cu(tmby)_2](TFSI)_2$ and $[Cu(tmby)_2](TFSI)$ as a HTM and employing the optimal mesoscopic $TiO_2$ scaffold (3.5 + 3 µm-thickness, optimization shown in Supplementary Table 1) as a working electrode under standard AM1.5G conditions. We see that our ssDSCs are superior to the state-of-the-art counterparts with various HTMs[7–12]. The histograms of $J_{sc}$, $V_{oc}$ and FF are presented in Supplementary Fig. 10. Compared to these HTMs, the $[Cu(tmby)_2]^{2+/+}$ for ssDSCs demonstrates an average PCE of 10.2% yielded by the average $J_{sc}$, $V_{oc}$ and FF of 13.70 mA cm$^{-2}$, 1.07 V and 0.696, respectively. The average $V_{oc}$ of ssDSCs is the same as that of liquid DSCs, indicating that the HOMO of Cu(II/I) HTM is comparable to the redox potential of the Cu(II/I) measured by cyclic voltammetry[15].

As shown in Fig. 5b, our best ssDSC reaches a record 11.0% under standard AM1.5G radiation at 1,000 W m$^{-2}$, with $J_{sc} = 13.87$ mA cm$^{-2}$, $V_{oc} = 1.08$ V and FF = 0.733. At 500 W m$^{-2}$, the PCE achieves 11.3% with $J_{sc} = 7.00$ mA cm$^{-2}$, $V_{oc} = 1.06$ V and FF = 0.760 and the PCE attains 10.5% at 100 W m$^{-2}$, with $J_{sc} = 1.40$ mA cm$^{-2}$, $V_{oc} = 1.01$ V and FF = 0.746. Our ssDSC is hysteresis free regardless of the scan rate, which ranges from 20 to 500 mV s$^{-1}$, as shown in Supplementary Fig. 11. Figure 5c presents the incident monochromatic photon-to-electron conversion efficiency (IPCE). The maxima of the IPCE is below 90% mainly due to the quenching of photoexcited dye molecules by copper complexes[38] in the HTM, charge recombination and optical loss (such as reflection and transmission, parasitic absorption of the HTM). The $J_{sc}$ calculated from the overlap integral of the IPCE with the standard AM1.5G emission spectrum (American Society for Testing and Materials G173-03) is 13.44 mA cm$^{-2}$, a value comparable to the $J_{sc}$ obtained under simulated AM1.5G conditions. We evaluate the photostability of a ssDSC operating at maximum output power for 200 h under radiation at 500 W m$^{-2}$, as shown in Fig. 5d. Remarkably, the $P_{max}$ retains over 85% of its initial value after 200 h due to slight improvement in $J_{max}$ and a small loss in $V_{max}$.

## Discussion

We demonstrate a 11% efficiency stable ssDSC under AM1.5G standard using a solid HTM composed of a blend of $[Cu(tmby)_2](TFSI)_2$ and $[Cu(tmby)_2](TFSI)$ and infiltrated in a 6.5 µm thick mesoscopic $TiO_2$ scaffold sensitized with Y123. This PCE is the highest achieved so far in the ssDSCs compared to previously reported HTMs and it is achieved by simultaneously high $J_{sc}$ of 13.87 mA cm$^{-2}$ and a high $V_{oc}$ of 1.08 V. The high

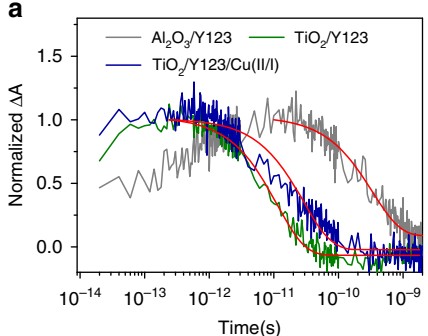
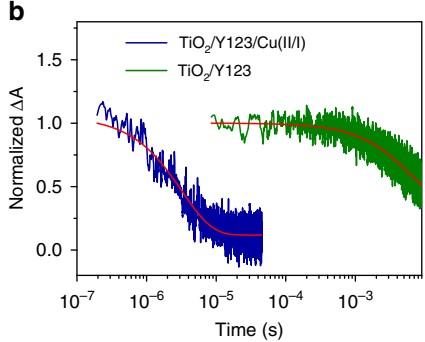

**Figure 4 | Time-resolved laser spectroscopy of interfacial electron transfer involving Y123 dye molecules.** (**a**) Transient absorption traces were probed at 650 nm following femtosecond laser pulsed excitation at 550 nm. (**b**) Transient absorption traces were probed at 715 nm following nanosecond laser pulsed excitation at 532 nm. Samples: $Al_2O_3$/Y123/inert electrolyte (grey); $TiO_2$/Y123/inert electrolyte (olive); $TiO_2$/Y123/Cu(II/I) hole conductor (royal blue). The inert electrolyte contains 0.1 M lithium bis(trifluoromethylsulfonyl)imide and 0.6 M TBP in acetonitrile. The red lines are monoexponential fits of the data.

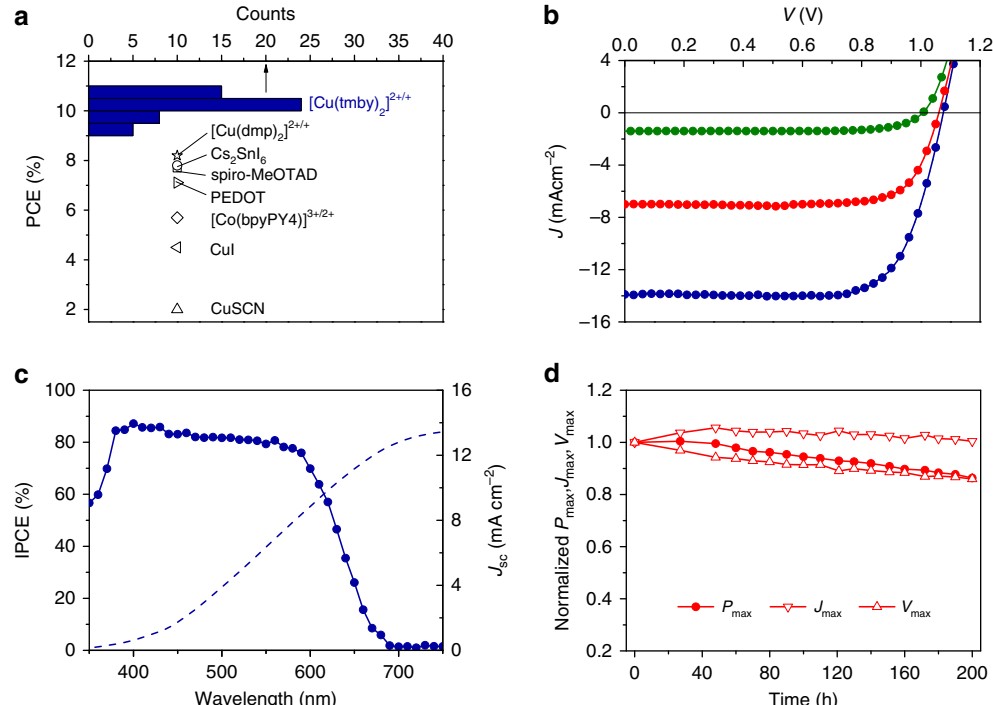

**Figure 5 | Performance of solid-state DSCs with Cu(II/I) hole conductor.** (**a**) Histogram of PCE of our ssDSCs based on $[Cu(tmby)_2]^{2+/+}$ as a hole conductor, compared to the reported PCEs of efficient ssDSCs using copper(I) thiocyanate, copper(I) iodide, $[Co(bpyPY4)]^{3+/2+}$, PEDOT, spiro-MeOTAD, $Cs_2SnI_6$ or $[Cu(dmp)_2]^{2+/+}$ as hole conductors. (**b**) The $J$–$V$ curves of a champion ssDSCs under standard AM1.5G radiation at 1,000 (royal blue), 500 (red) and 100 W m$^{-2}$ (olive). (**c**) IPCE spectrum and $J_{sc}$ calculated from the overlap integral of the IPCE with the standard AM1.5G emission spectrum (American Society for Testing and Materials G173-03). (**d**) Evolution of normalized $P_{max}$, $J_{max}$ and $V_{max}$ of our ssDSC operating at output maximum power under radiation at 500 W m$^{-2}$.

performances are mainly ascribed to the efficient photon harvesting by using the thick $TiO_2$ scaffold, favourable energetic alignment and efficient charge separation at $TiO_2$/Y123/HTM interfaces and rapid hole hopping in the amorphous HTM. Our work will foster future developments of low-cost hybrid photovoltaic such as sensitized solar cells and perovskite solar cells[39] based on solid-state metal complexes HTMs.

## Methods

**Materials.** Acetonitrile (ABCR), *t*-butanol (Sigma-Aldrich), LiTFSI (TCI) and TBP (TCI) were purchased from commercial company and used as received, unless stated otherwise. The $[Cu(tmby)_2]$(TFSI) and $[Cu(tmby)_2]$(TFSI)$_2$ powders were synthesized as previously described in the literature[15]. The $[Cu(tmby)_2]$(TFSI)$_2$ powders contain chloride anion impurity at p.p.m.

**Fabrication of solar cells.** The fabrication of DSCs followed the literature procedure[25]. Briefly, the 6.5 μm-thick mesoporous $TiO_2$ films (3.5 μm transparent layer + 3.0 μm light scattering layer) were immersed in 100 μM Y123 solution in ACN/*t*-butanol (v/v, 1/1) for 16 h to graft the dye molecules onto the $TiO_2$ surface followed by rinsing with ACN and drying with nitrogen flow. The counter electrodes consisted of PEDOT films electrochemically deposited on FTO glass[15]. The dye-coated $TiO_2$ working electrode and the counter electrode were assembled by using thermoplastic spacer (Surlyn, DuPont) heating at 120 °C. Electrolytes were injected into the space between the electrodes through predrilled hole on the counter electrode. The hole was sealed by using the thermoplastic sheet and a glass cover. The ssDSCs were obtained by removing the sealing on the hole to evaporate solvents in ambient air.

**Characterization of solar cells.** Solar cells were used without antireflection films and masked to an aperture area of 0.158 cm$^2$ for $J$–$V$ and IPCE characterizations. $J$–$V$ characteristics were recorded by a Keithley 2400 source meter. The solar cells were measured under radiation at 1,000 W m$^{-2}$ provided by a 450 W Xenon lamp of the Oriel solar simulator. The Oriel is equipped with a SchottK113 Tempax sunlight filter (Praezisions Glas & OptikGmbH) to match the emission spectrum of the lamp to the AM1.5G standard. The light intensity was determined using a calibrated Si reference diode equipped with an infrared cutoff filter (KG-3, Schott). IPCE spectra were measured with a lock-in amplifier (Stanford Research System

SR830 DSP) under chopped monochromatic light (2 Hz) generated by white light source from a 300 W xenon lamp passing through a Gemini-180 double monochromator (Jobin Yvon Ltd). The solar cell is illuminated under a constant white light bias with intensity of 50 W m$^{-2}$ supplied by an array of white light-emitting diodes.

EIS measurements were performed with a BioLogic SP300 potentiostat coupled with light-emitting diode light array to provide white light radiation at 1,000 W m$^{-2}$ onto DSCs. To obtain the spectra, the solar cell was biased with potentials and a modulation of 15 mV in the frequency range (1 MHz–0.1 Hz). Z-view software (v2.8b, Scribner Associates Inc.) was used to analyse the impedance spectroscopy. The equivalent circuit models for EIS analysis and the calculation of conductivity are presented in the Supplementary Information.

**SEM characterization.** SEM and energy dispersive X-ray spectroscopy (EDX) were carried out on a MERLIN high-resolution SEM (Zeiss, Germany). The sample for SEM measurement was obtained by removing the PEDOT counter electrode of a ssDSC.

**XRD.** X-ray powder diffractions were recorded on an X'Pert MPD PRO (Panalytical) equipped with a ceramic tube (Cu anode, $\lambda = 1.54060$ Å), a secondary graphite (002) monochromator and a RTMS X'Celerator (Panalytical) in an angle range of $2\theta = 5$–$60°$. Samples for XRD measurements were obtained by removing the PEDOT counter electrode of ssDSCs.

**Steady-state UV/Vis and photoluminescence spectroscopy.** UV/Vis absorption data was collected by a Perkin-Elmer Lambda 950 spectrophotometer and the extinction coefficients were calculated using the Beer-Lambert law. Steady-state photoluminescence spectra were recorded by exciting the samples at 450 nm with a 450-W Xenon CW lamp. The signal was recorded with a spectrofluorometer (Fluorolog; Horiba Jobin Yvon Technology FL1065).

**Time-correlated single photon counting.** The time-correlated single photon counting measurements were performed by exciting samples with a laser source at 408 nm (Horiba NanoLED 402-LH; pulse width less than 200 ps, 11 pJ per pulse, ~1 mm$^2$ in spot size) to generate a train of excitation pulses at 10 MHz. Decay curves were analysed with the software DAS-6 and DataStation provided by Horiba Jobin Yvon.

**Pump-probe spectroscopy.** Ultrafast transient absorbance spectra were acquired using pump-probe spectroscopy. A chirped pulse-amplified Ti:Sapphire laser (CPA-2001, Clark-MXR, 778 nm fundamental central wavelength, 120 fs pulse duration, 1 kHz repetition rate) was pumping a two-stage non-colinear optical parametric amplifier to obtain the pump beam at $\lambda_{ex} = 550$ nm. The pump fluence was of $25 \,\mu J \, cm^{-2}$ at the sample. The probe beam was generated in a $CaF_2$ crystal, yielding a white light continuum across the visible splitted in a signal and a reference beams that were directed to spectrographs (Princeton Instruments, Spectra Pro 2150i) and detected pulse-to-pulse with $512 \times 58$ pixels back-thinned CCD detectors (Hamamatsu S07030-0906). Motorized translation stage on the pump path controlled the acquisition of transient traces. The pump beam was chopped at half of the laser frequency and we typically averaged 3,000 laser shots to obtain satisfactory signal-to-noise ratio.

Samples for transient absorption spectroscopy were excited at 532 nm using an Ekspla NT-342 Q-switched Nd:YAG laser (pulse width: 4–5 ns; repetition rate: 20 Hz) and probed at 715 nm using a monochromator coupled with the probe light source of halogen lamp. The sample was positioned at $\sim 45°$ angle with respect to the incoming laser pulse. The signal was detected using the photomultiplier tube R9110 from Hamamatsu and recorded using the oscilloscope DPO 7254 from Tektronix. The output power density of the laser was attenuated using grey optical density filters to $50 \,\mu J \, cm^{-2}$. An acquisition was averaged over 3,000 laser shots.

**Data availability.** The data that support the findings of this study are available from the corresponding author on request.

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

## Acknowledgements

We acknowledge financial support from Swiss National Science Foundation (grant no. 200021-157135/1 and NCCR MUST research instrument) and CTI 17622.1 PFNM-NM, glass2energy SA (g2e), Villaz-St-Pierre, Switzerland. We thank Paul Liska, Robin Humphry-Baker, and Jean-David Decoppet for technical supports.

## Author contributions

M.G. and A.H. supervised the study. Y.C., Y.S. and M.F. devised the experiments. Y.C. fabricated and characterized the solar cells and wrote the manuscript with revisions from co-authors. Y.C., Y.S., A.U. and M.F. performed and analysed the absorption and emission spectroscopy, TCSPC and XRD experiments. Y.S. and M.F. synthesized copper complexes powder and made PEDOT counter electrodes. J.T. and J.-E.M. designed, performed and analysed the transient spectroscopy experiments. J.L. carried out SEM and EDS measurements. N.P. performed photostability test. F.G. contributed to the EIS measurements and analysis. S.M.Z. coordinated the work.

**Additional information**

**Competing interests:** The authors declare no competing financial interests.

