## [Peer Review File · Nature Communications]

Reviewers' comments:

Reviewer #1 (Remarks to the Author):

The manuscript shows record efficiency devices for solid state DSSCs using Cu(II/I) complexes as part of the hole transporting material. Compared to the previous publication by some of the authors (Ref. 6), this work shows that device efficiency can be boosted using a different structure for the Cu based HTM. The way Cu complexes design influences performance of DSSCs was reported in ref. 15, which discusses influence of the complex structure and its redox level on the regeneration kinetics. The paper also includes some structural analysis and measurements of the kinetics of charge transfer at the TiO₂/dye/HTM interface. The study is in principle suitable for publication in Nat. Comm.

However, there are a number of comments the authors should address which might involve providing some further characterization of the devices in the manuscript.

1) The authors suggest that the improved conductivity of the HTM is crucial to the improved performance of the device with respect to the liquid electrolyte counter part. However, close to short circuit conditions it has been shown that electron transport in TiO₂ can be limiting charge collection efficiency (see for example Leijtens et al. Adv. Mat. 2013). In Ref. 6 the authors themselves comment that transport in TiO₂ improves when going from liquid to solid state HTMs. Can the authors clarify this point?

2) The authors emphasize that slow evaporation of the solvent is needed to achieve high performance. In addition they comment on crystallinity of the HTM as a detrimental factor to the conductivity of the HTM. Could the authors comment on what the relation between evaporation rate of the solvent and crystallinity of the HTM is? In the section "Crystallinity of solid Cu(II/I) HTM" the light intensity dependence of two devices showing different HTM crystallinity is compared but there is no reference to differences in preparation method. A similar comparison but for different solvent evaporation rate might clarify some of these issues, assuming this results in more pronounced differences in HTM crystallinity, HTM conductivity, j_{sc} vs light intensity.

3) Apart from PEDOT, have the authors tried other contacts?

4) It is not clear to me how the connection between the HTM and the PEDOT counter electrode is guaranteed? Evaporation of the solvent must result in voids within the ~10µm spacing between the TiO₂ and the counter electrode? Have the authors quantified this spacing filling?

5) Regarding the TAS measurements, repetition rates of 1kHz and 20 Hz have been used for the electron injection study and the dye regeneration study respectively. Electron concentrations in samples with or without the HTM will be very different for the ultrafast measurements at this repetition rate (given that electron dye recombination happens in the millisecond timescale), which could also be causing changes in injection. A comment about this is needed. The analysis of the kinetics of regeneration might suffer from a similar problem though to a lower extent.

Also, have the authors tried TAS measurements on samples with crystalline versus amorphous HTM. Is there any difference in regeneration kinetics?

6) Is there a way to monitor the actual amount of solvent left in the cell during evaporation and correlate this quantity to the variation of the performance in time?

7) Are the SEM images taken on samples made without the Counter electrode or on samples made with counter electrode which is then removed? In the second case, a comment on how the HTM structure is expected to be different in the two cases is needed.

8) What do the authors think the role of the additives is (LiTFSI)? the complex is effectively already doped, how do cells without LiTFSI perform?

9) It is not clear how much the processing technique is crucial to the high performance of the cell. Have the authors tried to fabricate a device using the same procedure but using spiro OMeTAD dissolved in chlorobenzene, could this method improve the pore filling performance of other HTM? On the other hand, how does a solar cell perform when the Cu complex is spin coated onto the dyed TiO₂ layer?

10) The small reorganization energy of the complex is mentioned a number of times as the reason for better transport. Could the authors elaborate on the energy level matching and reorganization

energy of the dyes regarding regeneration? The presence/absence of the electrolyte does not seem to make a big difference for the regeneration kinetics (TAS data) when compared to data in ref. 15.

11) I assume what the authors call "discussion" are the conclusions of the manuscript

There are several typos throughout the manuscript and SI.

Reviewer #2 (Remarks to the Author):

The manuscript describes significant advances in the field of solid state dye sensitized solar cells, a technology that has seen few improvements over the last years. The work here, however, marks a milestone for the technology: achieving >10% PCEs. The authors do so by preparing liquid electrolyte DSCs and letting the ACN solvent evaporate out of the device, leaving behind solid state copper complexes with remarkable good properties. The devices also appear to show promising stability. As a result, the impact of the work is high and it should be published in nature communications. However, there are a few minor issues to be addressed:

1. The authors make the ssDSCs by allowing the solvent to slowly evaporate from the cells. Can the authors not make ssDSCs via a process more compatible with a solid state solar cell such as printing and annealing? Can a metal electrode be evaporated on top of the dried samples for improved contact? How important is the fact that the electrolyte leaves slowly over many days?
2. On this note, the performance appears to be steadily dropping over time after the first few days. What is causing this decrease in JSC? Do the devices function best with a very small amount of ACN and charge transport is hindered as it completely dries? Is the crystallinity of the HTM changing over time?
3. The experimental section describes the use of a PEDOT layer on the FTO counterelectrode but this is not mentioned in the main text. How important is this pedot layer? As the ACN evaporates, how does the contact and morphology between the dried HTM and the counterelectrode change? Would not a conformal evaporated or sputtered electrode form better contact?
4. The authors show that the HTM absorbs considerably in the visible part of the spectrum, which must result in parasitic absorption. This could be part of the origin for the low EQE values.
5. The best DSCs use multiple dyes to effectively absorb throughout the visible and even NIR spectrum. Can the TiO_2 be co-sensitized in this system, or are the energy levels of the HTM not suitable for a smaller bandgap dye?
6. The energy levels of the redox system in liquid and solid state should be measured and reported. This is extremely important.

Reviewer #3 (Remarks to the Author):

This paper demonstrates the potential to further optimize the performance of a ssDSC solar cell by improving the HTM hole transport layer properties using newly develop amorphous Cu(II/I) compounds. In addition, the authors have chosen the cell layers to achieve a V_{oc} over 1.0 eV.

This paper is well written and explains clearly the static and dynamics of the cell operation. I suggest that the paper be published after the authors have a chance to include the following:

1. It would be helpful to the general readers if the authors can include in the paper the band alignment of the layers for the cell to show how the high value of the V_{oc} comes about.
2. How did the authors go about to optimize the thickness of the porous TiO_2 layer? The results are impressive, but can one not push the thickness further to increase the efficiency?

Reviewers' comments:

Reviewer #1 (Remarks to the Author):

The manuscript shows record efficiency devices for solid state DSSCs using Cu(II/I) complexes as part of the hole transporting material. Compared to the previous publication by some of the authors (Ref. 6), this work shows that device efficiency can be boosted using a different structure for the Cu based HTM. The way Cu complexes design influences performance of DSSCs was reported in ref. 15, which discusses influence of the complex structure and its redox level on the regeneration kinetics. The paper also includes some structural analysis and measurements of the kinetics of charge transfer at the TiO₂/dye/HTM interface. The study is in principle suitable for publication in Nat. Comm.

However, there are a number of comments the authors should address which might involve providing some further characterization of the devices in the manuscript.

1) The authors suggest that the improved conductivity of the HTM is crucial to the improved performance of the device with respect to the liquid electrolyte counter part. However, close to short circuit conditions it has been shown that electron transport in TiO₂ can be limiting charge collection efficiency (see for example Leijtens et al. Adv. Mat. 2013). In Ref. 6 the authors themselves comment that transport in TiO₂ improves when going from liquid to solid state HTMs. Can the authors clarify this point?

Answer: The paper (Leijtens et al. Adv. Mat. 2013) shows that at low charge densities ($< 2 \times 10^{17} \text{cm}^{-3}$) under short circuit conditions, low TiO₂ conductivity will limit charge conduction in solar cells. Note that this is irrelevant for the device performance since the photocurrents and hence the IR loss are very small under these conditions. However, at high charge densities ($> 2 \times 10^{17} \text{cm}^{-3}$), a strong increase in TiO₂ conductivity results in the HTM becoming the limiting component in charge transport. Using charge extraction measurements, we found that the carrier density is over $7 \times 10^{17} \text{cm}^{-3}$ in our solar cells under one Sun, indicating that here the charge transport is limited by the HTM. Therefore, improving the conductivity of the HTM can improve the charge transport and the J_{sc} as well as the FF. We have now added this discussion in the manuscript.

2) The authors emphasize that slow evaporation of the solvent is needed to achieve high performance. In addition they comment on crystallinity of the HTM as a detrimental factor to the conductivity of the HTM. Could the authors comment on what the relation between evaporation rate of the solvent and crystallinity of the HTM is? In the section "Crystallinity of solid Cu(II/I) HTM" the light intensity dependence of two devices showing different HTM crystallinity is compared but there is no reference to differences in preparation method. A similar comparison but for different solvent evaporation rate might clarify some of these issues, assuming this results in more pronounced differences in HTM crystallinity, HTM conductivity, j_{sc} vs light intensity.

Answer: A high rate of solvent evaporation leads to the rapid crystallization of the resulting in poor contact of HTM with the electrodes. In the Supporting Information, we have now added an SEM image and XRD data (see Supplementary Figure 2) of a sample with the architecture FTO/TiO₂/Y123/HTM made by a rapid solvent evaporation. The SEM image shows the poor contact of the HTM with the electrodes, which we posit leads to poor carrier collection and hence low performance of these DSSCs.

3) Apart from PEDOT, have the authors tried other contacts?

Answer: Apart from PEDOT, we tried low-cost counter electrodes such as graphene-coated FTO (doi:10.1038/nchem.1861). The ssDSSCs with the graphene counter electrode have average PCE = $7.9 \pm 0.2\%$, $J_{sc} = 11.85 \pm 0.44 \text{ mA/cm}^2$, $V_{oc} = 1.07 \pm 0.01 \text{ V}$, FF = 0.62 ± 0.01 , which are lower than the devices with PEDOT counter electrode shown in the present manuscript.

4) It is not clear to me how the connection between the HTM and the PEDOT counter electrode is guaranteed? Evaporation of the solvent must result in voids within the

~10um spacing between the TiO₂ and the counter electrode? Have the authors quantified this spacing filling?

Answer: We used the thermoplastic spacer with proper size to leave a space around the TiO₂ film filled with the liquid electrolyte. During the evaporation of solvent, the void between the two electrodes can be filled by the electrolyte located around the TiO₂ film.

5) Regarding the TAS measurements, reprints of 1kHz and 20 Hz have been used for the electron injection study and the dye regeneration study respectively. Electron concentrations in samples with or without the HTM will be very different for the ultrafast measurements at this reprints (given that electron dye recombination happens in the millisecond timescale), which could also be causing changes in injection. A comment about this is needed. The analysis of the kinetics of regeneration might suffer from a similar problem though to a lower extent. Also, have the authors tried TAS measurements on samples with crystalline versus amorphous HTM. Is there any difference in regeneration kinetics?

Answer: The electron density in TiO₂ could affect injection, but we are at a very low light regime, ensuring low electron concentration in the nano-pico second time. We have now added this discussion to the manuscript. Regeneration at 20Hz is safe, with 50ms for the sample to recover.

We did not try TAS measurements on samples with amorphous and crystalline HTM. We believe it goes beyond the scope of the present manuscript.

6) Is there a way to monitor the actual amount of solvent left in the cell during evaporation and correlate this quantity to the variation of the performance in time?

Answer: The solvent evaporation from unsealed solar cells is processed in ambient air. The rate of solvent evaporation can be sensitive to the ambient conditions (temperature and pressure). It's technically difficult to monitor the rate of solvent evaporation or the actual amount of solvent left in the cell during evaporation. Nevertheless, we believe that keeping the cells in ambient air for a considerable period of time as we do for our best-performing cells, would result in nearly complete solvent evaporation. The solid Cu(I/II) complex I left behind is clearly visible in the SEM pictures presented.

To check on the effect of residual solvent we kept our unsealed devices under high vacuum provide by an oil pump for one hour to withdraw any residual solvent. We find that the device performance remains stable (before vacuum: $J_{sc}=13.10\pm0.78$ mA/cm², $V_{oc}=1.07\pm0.01$ V, FF=0.68±0.03 and PCE=9.5±0.2%; after vacuum: $J_{sc}=13.24\pm0.58$ mA/cm², $V_{oc}=1.08\pm0.01$ V, FF=0.68±0.03 and PCE=9.6±0.2%). This result indicates that a small amount of residual solvent has negligible effect on cell performance.

7) Are the SEM images taken on samples made without the Counter electrode or on samples made with counter electrode which is then removed? In the second case, a comment on how the HTM structure is expected to be different in the two cases is needed.

Answer: The SEM images were taken on samples made with counter electrode which is then removed. We have now clarified this aspect and mentioned it in the Methods.

8) What do the authors think the role of the additives is (LiTFSI)? the complex is effectively already doped, how do cells without LiTFSI perform?

Answer: We fabricated ssDSC with and without LiTFSI in the HTM. The device with LiTFSI has $J_{sc}=13.35$ mA/cm², $V_{oc}=1.08$ V, FF=0.68 and PCE=9.9%. However, the device without LiTFSI has $J_{sc}=11.48$ mA/cm², $V_{oc}=1.08$ V, FF=0.69 and PCE=8.5%. Thus the presence of lithium salt improves the J_{sc} by about 20% without affecting the other PV metrics. Note that Leitens et al claim that for their HTM, Li doping is not crucial (Leijtens et al. Adv. Mat. 2013). The role of lithium salt in DSCs is not completely clear. We think that the lithium salt in our ssDSC could tune the energetic and kinetic properties at TiO₂/dye/HTM interfaces (DOI:10.1002/adfm.201100048). It may also inhibit crystallization of the Cu complex.

9) It is not clear how much the processing technique is crucial to the high performance of the cell. Have the authors tried to fabricate a device using the same procedure but using

spiro OMeTAD dissolved in chlorobenzene, could this method improve the pore filling performance of other HTM? On the other hand, how does a solar cell perform when the Cu complex is spin coated onto the dyed TiO₂ layer?

Answer: The processing technique in our work allows the use of 6.5 μm thick mesoscopic TiO₂ film to improve light harvesting efficiency, guarantees the homogenous infiltration of the HTM homogeneously in the thick TiO₂ scaffold and yields amorphous HTM. We did not fabricate a device using spiro-OMeTAD with the same method, considering the low vapor pressure of chlorobenzene.

We also did not make a solar cell based on Cu(II)/Cu(I) spin-coated onto TiO₂ film, because fast solvent evaporation can lead to poor contact of the crystalline HTM with the electrodes as mentioned in the response to question 2. The formation of crystalline phase of the copper complex when a spin-coating procedure was used has already been shown in a previous study (Ref. 6, DOI:10.1039/C5EE01204J), which we cite.

10) The small reorganization energy of the complex is mentioned a number of times as the reason for better transport. Could the authors elaborate on the energy level matching and reorganization energy of the dyes regarding regeneration? The presence/absence of the electrolyte does not seem to make a big difference for the regeneration kinetics (TAS data) when compared to data in ref. 15.

Answer: We have now added a diagram of the energy levels in our device to elaborate on the energy level matching (see Supplementary Figure 1) and commented on the role of the low reorganization energy in enabling rapid regeneration of the sensitizer with a minimal driving force

The driving force for dye regeneration is around 0.12 V in liquid DSCs. The HOMO of solid HTM could be comparable to the redox potential of Cu(II/I), considering that the average V_{oc} of ssDSCs is comparable to that of liquid DSCs. Therefore, the driving forces for dye regeneration in ssDSCs and liquid DSCs are similar, which could result in comparable dye regeneration kinetics.

11) I assume what the authors call "discussion" are the conclusions of the manuscript

Answer: Indeed and this has been corrected.

There are several typos throughout the manuscript and SI.

Answer: Many thanks for the note. We have now corrected the typos in the manuscript and Supporting Information.

Reviewer #2 (Remarks to the Author):

The manuscript describes significant advances in the field of solid state dye sensitized solar cells, a technology that has seen few improvements over the last years. The work here, however, marks a milestone for the technology: achieving >10% PCEs. The authors do so by preparing liquid electrolyte DSCs and letting the ACN solvent evaporate out of the device, leaving behind solid state copper complexes with remarkable good properties. The devices also appear to show promising stability. As a result, the impact of the work is high and it should be published in nature communications. However, there are a few minor issues to be addressed:

1. The authors make the ssDSCs by allowing the solvent to slowly evaporate from the cells. Can the authors not make ssDSCs via a process more compatible with a solid state solar cell such as printing and annealing? Can a metal electrode be evaporated on top of the dried samples for improved contact? How important is the fact that the electrolyte leaves slowly over many days?

Answer: We did not make devices via printing and annealing as these processes could induce the growth of large crystals of the HTM, leading to low device performance. The

slow evaporation of solvents in the electrolyte is crucial to obtaining an amorphous HTM in good contact with the electrodes.

2. On this note, the performance appears to be steadily dropping over time after the first few days. What is causing this decrease in J_{sc}? Do the devices function best with a very small amount of ACN and charge transport is hindered as it completely dries? Is the crystallinity of the HTM changing over time?

Answer: We presented the variations of J_{sc} in the Figure 1c. The highest J_{sc} can be stable for 20 days. However, the J_{sc} of a few devices showed a marked decrease, reducing the average J_{sc} and leading to large variations. We reason that the drop of J_{sc} is due to the formation of crystalline HTM, causing sublinear dependence of J_{sc} on light intensity. The fact that the drop-in photocurrent is related to decrease in carrier collection due to loss of electric contact is supported by the fact that the photocurrent measured at low light intensity remained stable.

The average PCE of devices reaches a peak value after a week. We tried to keep our unsealed devices under vacuum (by oil pump) for one hour, and the device performance remains stable (before vacuum: J_{sc}=13.10±0.78 mA/cm², V_{oc}=1.07±0.01 V, FF=0.68±0.03 and PCE=9.5±0.2%; after vacuum: J_{sc}=13.24±0.58 mA/cm², V_{oc}=1.08±0.01 V, FF=0.68±0.03 and PCE=9.6±0.2%). This result indicates that a very small amount of ACN has negligible effects on the best performing cells.

3. The experimental section describes the use of a PEDOT layer on the FTO counterelectrode but this is not mentioned in the main text. How important is this pedot layer? As the ACN evaporates, how does the contact and morphology between the dried HTM and the counterelectrode change? Would not a conformal evaporated or sputtered electrode form better contact?

Answer: We have now added the description of the use of a PEDOT as counter electrode in the manuscript. The PEDOT layer is mesoporous (*Electrochimica Acta* (2013), 107, 45, <http://dx.doi.org/10.1016/j.electacta.2013.06.005>), offering a large surface area contact and probably high catalytic activity with the HTM. We tried low-cost graphene-coated FTO counter electrode (doi:10.1038/nchem.1861), which yields lower device performance (average PCE=7.9±0.2%, J_{sc}=11.85±0.44 mA/cm², V_{oc}=1.07±0.01 V, FF=0.62±0.01) than the PEDOT counter electrode. It remains to be checked whether a thermally evaporated or sputtered electrode could form a better contact. This method requires the HTM to be spin-coated on the electrode, which leads to poor device performance as described in the response of question 1.

4. The authors show that the HTM absorbs considerably in the visible part of the spectrum, which must result in parasitic absorption. This could be part of the origin for the low EQE values.

Answer: In the main manuscript, we have shown that the dye molecule Y123 has a much higher molar extinction coefficient than copper complexes in the visible part of the spectrum, indicating most of photons harvested by Y123. The origin for low EQE can be attributed mainly to the quenching, recombination and optical loss as described in the manuscript.

5. The best DSCs use multiple dyes to effectively absorb throughout the visible and even NIR spectrum. Can the tio2 be co-sensitized in this system, or are the energy levels of the HTM not suitable for a smaller bandgap dye?

Answer: Co-sensitized TiO₂ with smaller bandgap dye would work in this system, if the LUMO and HOMO of the dye matches with the energy levels of TiO₂ and the copper complex HTM, respectively. We agree with the referee that this would be interesting and we believe such a study is an excellent idea for future research directions with our HTM.

6. The energy levels of the redox system in liquid and solid state should be measured and reported. This is extremely important.

Answer: The redox potential of the Cu(tmby)₂ in liquid state measured by cyclic voltammetry method was reported in our previous work (Ref. 15, DOI:10.1021/jacs.6b10721). We have now included this value in the manuscript for

clarification. We also show an energy level diagram in Figure 1 of the supplementary section. The HOMO of solid state Cu(II/I) HTM should be comparable to the redox potential of the Cu(tmby)₂ measured by the cyclic voltammetry method, as the average V_{oc} of ssDSCs is comparable to that of the liquid DSCs. We have now described this aspect in the manuscript.

Reviewer #3 (Remarks to the Author):

This paper demonstrates the potential to further optimize the performance of a ssDSC solar cell by improving the HTM hole transport layer properties using newly developed amorphous Cu(II/I) compounds. In addition, the authors have chosen the cell layers to achieve a V_{oc} over 1.0 eV.

This paper is well written and explains clearly the static and dynamics of the cell operation. I suggest that the paper be published after the authors have a chance to include the following:

1. It would be helpful to the general readers if the authors can include in the paper the band alignment of the layers for the cell to show how the high value of the V_{oc} comes about.

Answer: We have now included a diagram of energy levels in our DSCs (see Supplementary Figure 1) and elaborated on the origin of high V_{oc} in the manuscript.

2. How did the authors go about to optimize the thickness of the porous TiO₂ layer? The results are impressive, but can one not push the thickness further to increase the efficiency?

Answer: The ssDSCs presented in the manuscript have the optimal porous TiO₂ layer. We have now included the photovoltaic results of Cu(II/I) based ssDSCs with different thickness of TiO₂ films in the Supplementary Table 1, where we demonstrate how the optimal thickness was identified.

Reviewers' comments:

Reviewer #2 (Remarks to the Author):

The authors have responded to all of my comments, and so the paper can be published. There are still one or two small clarifications that could be made, however:

1. in their response regarding the charge density dependence of the TiO₂ conductivity, the authors claim that their currents are small and they hence have no IR loss. This may be true, but the losses induced by resistive TiO₂ are not just in the form of IR losses; slow electron transport also results in reduced collection efficiency and hence lower photocurrents. Are the charge density measurements of 10^{17} cm^{-3} at OC or at SC? these would have to be at SC to be relevant to the discussion of charge collection.

2. The authors claim that the dye absorbs more strongly than the HTM in the visible range. This is also the case for Spiro OMETAD, but cells with Spiro still suffer from large parasitic absorption. It would be worth quantifying the losses at some point to get an estimate of the maximum obtainable JSCs, using transfer matrix modeling with an effective medium approximation.

Reviewers' comments:

Reviewer #2 (Remarks to the Author):

The authors have responded to all of my comments, and so the paper can be published. There are still one or two small clarifications that could be made, however:

1. in their response regarding the charge density dependence of the TiO₂ conductivity, the authors claim that their currents are small and they hence have no IR loss. This may be true, but the losses induced by resistive TiO₂ are not just in the form of IR losses; slow electron transport also results in reduced collection efficiency and hence lower photocurrents. Are the charge density measurements of 10¹⁷ cm⁻³ at OC or at SC? these would have to be at SC to be relevant to the discussion of charge collection.

Answer: Indeed, under a low carrier density in TiO₂ (2x10¹⁷ cm⁻³) at short-circuit conditions, the slow electron transport can result in reduced collection efficiency and lower photocurrent. We now have clarified the charge density measurements at short-circuit conditions in the manuscript.

2. The authors claim that the dye absorbs more strongly than the HTM in the visible range. This is also the case for Spiro OMETAD, but cells with Spiro still suffer from large parasitic absorption. It would be worth quantifying the losses at some point to get an estimate of the maximum obtainable JSCs, using transfer matrix modeling with an effective medium approximation.

Answer: The ssDSCs based on spiro-OMeTAD generally use 2-3 um thick transparent mesoscopic TiO₂ films. If poor absorber such as Z907 was used, the solar cell indeed could suffer from large parasitic absorption loss (doi: DOI:10.1002/aenm.201300057). In our work, the ssDSCs based on Cu(II/I) HTM use mesoporous TiO₂ film composed of 3.5 um thick transparent layer and 3.0 um light scattering layer, together with a high absorption organic dye Y123. The parasitic absorption loss from the HTM can be alleviated.

We thank the referee for suggesting to estimate of maximum obtainable Jsc by using transfer matrix modeling. This profound study can be presented in a separate paper and the conclusion of the present work would be not affected without this study.

REVIEWERS' COMMENTS:

Reviewer #2 (Remarks to the Author):

The paper has addressed all concerns I had and should be published.